

# Enabling blockchain for Saudi Arabia drug supply chain using Internet of Things (IoT)

Saeed M. Alshahrani

Department of Computer Science, College of Computing and Information Technology, Shaqra University, Shaqra, Saudi Arabia

## ABSTRACT

The availability of drugs across the country is a direct measure for fairer public health. Several issues have been reported drastically related to various organizations that fail to provide quality medicines on time. There has been a consistent increase in cases where the treatment, as well as exempted drugs, were supplied due to the unavailability of proper traceability of the supply chain. Several parties are involved in the supply and have similar interests that may defer the adequate shareability of the drugs. The existing system for managing the drug supply chain suffers from several backlogs. The loss of information, unavailability of resources to track the proper medicinal storage, transparency of information sharing between various stakeholders and sequential access. The applicability of the decentralized model emerging from the blockchain can apply to one of the perfect solutions in this case. The drug traceability chain can be deployed to a Ledger-based blockchain that may result in decentralized information. Continuous supply from the Internet of Things (IoT) based devices might be handy as the middleware for providing a trustworthy, safe, and proper transaction-oriented system. The data integrity, along with the provenance resulting from the IoT-connected devices, is an effective solution towards managing the supply chain and drug traceability. This study presents a model that can provide a token-based blockchain that will help provide a cost-efficient and secure system for a reliable drug supply chain.

# INTRODUCTION

One of the most essential features in healthcare units across any nation is the eligibility to maintain a proper drug supply chain. To ensure the appropriate placement of trusted drugs that are not expired and are prepared with adequate delicacy, the Saudi Food and Drug Authority takes various measures. Despite the different security and safety measures the authorities took, there is a probable chance of human error that might lead to inappropriate logistic transfers of medicines or production drugs. Consumption of drugs which are not eligible to be eaten may lead to death. Several stakeholders are engaged in the pharmaceutical-level transfer of drugs.

## General information

All the stakeholders probably cannot take preventive measures, which might lead to embarrassing circumstances. To motivate the issue of security to produce such drugs that

Corresponding author
Saeed M. Alshahrani,
salshahrani@su.edu.sa

can save someone's life and to handle the complete situation based on the authorized and safe transmission of raw material to manufacture such drugs. The healthcare industry manages complex arrangements to ensure proper medications are available for patients in hospitals and medical centres. The connectivity between various stakeholders across the complete cycle is difficult to handle. Human efforts are required to ensure proper trust and security between supply chain members.

The supply chain process includes the manufacturing units for the drugs, distributors who purchase wholesale drugs, salesmen or medical representatives, pharmacies, and managers for the pharmaceutical supply. It is a sequential process that begins at the manufacturing level, and various resellers make the orders. The manufacturing agency must obtain special permissions and regulatory authorization from various national drug authorities. Before manufacturing starts, the stakeholders must get permission from the authorities, and special care must be taken regarding the manufacturing measures an organization adopts. By the end of the manufacturing cycle, these drugs are packed and loaded for delivery to the wholesale resellers. A distributor who purchases and places an order from these manufacturing units plays a vital logistical role in providing the supply chain with adequate measures to deliver the drop from the manufacturer site towards the pharmacy counters across the nation.

## Problem associated with medicinal supply chain

However, the complex nature of these organizational policies adopted by various stakeholders might lead to the introduction of several drugs that are not part of the manufacturing standards. These drugs may or may not be harmful to the customers. The consumption of such medicines in several countries includes 10–30% of counterfeit medications. These statistics show that the pharmacy industry is more likely to be a victim of counterfeiting drugs than average organizations. The stakeholders who are responsible for the delivery of the drugs may be victims of this vulnerability. The World Health Organization (WHO) stated that the significant reason for prevailing deaths in any developing nation is because of counterfeit drugs (*The Guardian, 2017*). They further noted that the primary victims of such medications are children. The transport of such medicines will also be used as an additional add-on to several problems related to environmental conditions. The exhaust fuel from the transport services will add to issues such as temperature, pressure, humidity, and pollution. It is worth mentioning that several drugs are sensitive to temperature variations, require special attention, and can be managed without handling in various parts of the country.

## Gaps between the technologies used

Several legacy measures that the conventional supply chain for drugs make use of databases or cloud-based services to control the process for the delivery of drugs from one place to another. In such systems, various stakeholders change the administration of the control units as per the stakeholders. Every stakeholder uses their admin privileges before the process drug reaches the consumer. The tracking of the product at an admin had moved or shipped from one place to another store, which stores the actual information,

which can be mutated at different times. The independent nature of information between various partners and the participants across the supply chain makes it difficult to locate the actual origin of the medicine. The data for medicinal usage involves various associated parties and several steps in the process. There is a low sense of coupling between several stakeholders, which may compromise the trust between multiple parties involved. A proper chain is required to overcome the Contra feeds problem and build trust between various entities involved in any pharmaceutical supply. Integrating technology is supposed to be one of the most amicable solutions. A peer-to-peer network that is compatible with providing a decentralized backbone of a network shared across multiple nodes to transact between various peers without a centralized control authority is called a blockchain. There is no owner for any mechanism, which results in the transparency of information flow in the blockchain. All the transactions that take place are under the governance of a timestamp, and the status of any transaction can be identified and located by all the parties at any given time. A subordinate use of the Internet of Things (IoT), sensor-based devices, tracking units and blockchain can result in the production of records that remain unaltered. All the shared transactions in the complete supply chain are monitored, and the proper tracking for these records is observed for the drugs that travel large distances from one point to another to save the lives of human beings. All information is available for stakeholders using Internet-based technologies such as blockchain. Several home care systems are designed under multiple studies to ensure the consumer about the authenticity and valid nature of the medicine (*Dammak et al., 2022*).

## Our contributions

This study shows a model related to the medical supply chain which uses these tokens to enable blockchain integration. To identify all the counterfeit drugs that might travel in the market due to a lack of proper tracking and traceability. Several contributors have different accessibility and systems to track their products. However, no transparent system can identify the needs of all the stakeholders. The engagement of various stakeholders for pharmaceutical drugs is the main focus of this research work. Our token-based, IoT-enabled supply chain ensures the active participation of older stakeholders. The actors presented in the entire chain are comprised of various units. The traceability of the drugs in the market is possible with the help of the NFT. These tokens carry complete information for the batch of drugs manufactured at an instance. Information related to the manufacturer, owner, manager, shipping, storage, billings, and other certificates related to the drug is mentioned. The blockchain for this supply chain comprises old information stated above. Medical history is accessible to the entire unit, and it becomes easy for the end-user to identify whether a drug is available at an instance or not. However, security remains a significant concern in the field of blockchain; the model proposed in this article comprises intelligent contracts established to utilise user access at different levels. The final comment of the blockchain contains all the information depending on the user's access, and information flowing from a specific user is updated based on his privileges. Smart contracts are a significant means for managing the privacy and security of the complete system. The implementation of the model, along with the validation and evaluation of

radius models, is done in this research. The information collection is done on a prototype basis with the help of a Raspberry Pie Model 4B integrated with several sensors and a GPS tracker. A local smart contract is deployed at the network to test the proposed model.

Depending on the associated business, a public or private blockchain can be deployed to handle pharmaceutical drugs. With the help of blockchain, a complete track record of the drugs from the manufacturing to the consumer is maintained. The movement of the drug and the information related to the movement can relate to the temperature associated with the existing transportation unit. Tracking the temperature and transport distance will help reduce the risk of counterfeiting drugs. The data access in the system can be done with the help of the exchange of intelligent contracts for the transactions. The integrity of the data collected from the IoT units and the supply chain's security is maintained with these smart contracts. The collected data from the consumer will be held for secrecy and privacy. Smart contracts in the supply chain ensure the execution of agreements between various parties and the transaction between stakeholders with reduced cost and enhanced safety (*Prause, 2019*). Non-fungible tokens (NFT) have recently gained popularity in various blockchain transaction mechanisms. These tokens are cryptography assets integrated into a blockchain with unique identification codes and material data to distinguish them from each other. These tokens represent commodities or goods in the form of regular tokenized items. This makes the transaction more straightforward to perform with less probability of making a fraud. These tokens can represent real-world assets, identities, writers, artwork, real estate, *etc*. The article presented in this study is presented in several sections. "Background" contains several concepts related to blockchain technology, smart contracts, and NFT tokens. A complete description of the existing work in this field is represented in "LIterature Review". "Proposed Model in the Study" gives a high-level overview of the utilization of traceability for medicine of those drugs. The implementation of the model described is explained in "Implementation". Further discussion and conclusion are provided in "Discussion" of this article.

## BACKGROUND

The first-ever blockchain was proposed by *Nakamoto (2008)* in which the author represented the transfer of digital funds. The most exciting feature of this study was the transparency and decentralization of the information. The first-ever blockchain was recognized and became one of the most important features for controlling finances and transparent data across various industries and organizations. The distributed ledger technology (*Puthal et al., 2018*) prescribed in the blockchains is supposed to be one of the most effective decentralized mechanisms for storing transactions. Since security is one of the significant concerns in blockchain-based transactions, using a hyper-ledger ensures that the data is transmitted with proper transparency of any user inputs. The process for validating the data committed in the blockchain is called mining. All the records related to mining are maintained by all the users in the supply chain. The blockchain serves as the unanimous privacy manager. Various nodes inside the blockchain contain sophisticated consensus mechanisms where the data flows from one node to another. The globally accessible ledger contains several information related to the transaction, including blocks,

user information, hash values, and data packet information. The information's tempering is difficult regarding the globally accessible block in the chain.

The resultant block, after tempering, will fail to ensure that a proper hash code is attached to the block. Ultimately, the block with any tempered situation is identified immediately and can be updated for further investigation. A combination of blockchain and smart contracts is required to avoid tempering and achieve a higher security rate. These smart contracts are algorithmic codes capable of executive transactions and operations. The use of such algorithms averts the need for any third-party involvement. One of the famous intelligent contracts execution is presented by *Wood (2014)* termed as Ethereum. The smart contracts were utterly implemented with the help of Solidity Language. Several researchers have compiled the integration of blocks in business systems along with decentralized data storage for secure and integral transactions. One of the implementations (*Nizamuddin, Hasan & Salah, 2018*) suggested IPFS blockchain for the authentication of digitally published information. The primary focus was on the publication of digital content online along with the integrity affirmation. Blockchain added to the power of integrity management, and intelligent contracts provided security for the content published online. Yet another research (*Hasan et al., 2020*) proposed the creation of an apparel clone of digital data to ensure traceability and transaction security. The storage of the data, along with the sharing, is done with the help of the IPFS paradigm. Implementing FileTribe proposed by *Sari & Sipos (2019)* uses smart contracts to share files across a closed user group. The authentication was done with the help of smart contracts in which a secure decentralized application was deployed. The use of IPFS in the system ensured the accessibility of the data in an intelligent way that was not dependent upon the centralized database approach. There are several more enhanced versions of smart contracts used in different systems by *Sultana et al. (2023)* (related to transactions) (*Dwivedi, Amin & Vollala, 2023*) (related to digital signatures) (*Alangari et al., 2022*) (related to contract security) and *Onwubiko et al. (2023)* (related to digital contract exchange). Unlike fungible tokens like Ether, non-fungible tokens play an essential role in security and privacy issues. The values stored within these tokens are integral, and they don't have a contrast with their existing peer tokens. They all have their unique identity and distinct nature, making them memorable to use in smart contracts. The uniqueness of the token was explained by *Wang et al. (2021)*. Integrating such tokens with the smart contracts and then with the blockchain makes it unique to ensure the owner's privacy and identity (*Turki et al., 2023*). The consistent nature of these tokens provides a unique identity to the assets which are a part of the supply chain mechanism. Several tokens of this type were used in models proposed by researchers like (*Vogelsteller & Buterin, 2015*). The author's Ethereum blockchain makes use of ERC-20 smart contracts. ERC 20 was discontinued because non-fungible tokens were not integrated into this standard. ERC 721 standard was equipped using NFT's (*Entriken et al., 2018*). Continuing the series for the tokens, a worldwide standard was designed that comprises the power of ERC 20 and ERC 721 features. ERC 1155 standard was given by *Radomski et al. (2018)* to handle both the categories of tokens required for secure and safe transactions.

# LITERATURE REVIEW

This literature review section focuses mainly on utilising blockchain and the tokenization approach to provide a safe and secure mechanism. *Almalki et al. (2022)* recommended that IoT can be integrated with healthcare in various possible ways. A model using non-fungible tokens integrated with the blockchain to provide a secure mechanism in the drug supply chain was proposed by *Turki et al. (2023)*. Recently, tokenisation and blockchain have been gaining popularity because tokens tend to enhance the security and safety of blockchain nodes. To facilitate all the transactions in the blockchain, the use of tokens can also improve the privacy of the nodes. Several research studies have been conducted regarding blockchain technology to provide secure supply chain integration. The traceability of several products manufactured in the organization was tracked with the help of IPFS and Ethereum, presented in the study by *Musamih et al. (2021)*. A proper systematic review was given by *Alshahrani et al. (2023)* on using Artificial Intelligence and its allied systems. The system's reliability is supposed to be enhanced significantly with an IPFS file storage system. As per the review from *Sargent & Breese (2024)*, proposed challenges in several supply chains. The ability to collect data is also secured in this type of approach. Smart contracts are created for every movement of data in the blockchain to ensure a clean transaction. A medical ledger was implemented in a study given by *Mattke et al. (2019)*. This article suggested a pharmaceutical drug supply chain in which the spread of counterfeit drugs was minimized significantly. The model achieved control over the drugs being supplied, along with a decrease in counterfeiting. Yet another model was designed by *Jamil et al. (2019)* in which the information for all the stakeholders was filled inside the blockchain. Tracking the information for older stakeholders where the transactions occurred between two parties was easy. The exchange of smart contracts takes place with legal and legitimate users only. Tracking the transactions was one of the most exciting features of this study. However, the system's output was measured directly with the latency and transaction processing time. A hyper ledger fabric was used in one of the researchers posted by *Azzi, Chamoun & Sokhn (2019)* in which the stakeholders did the authorization to upload any data or transactions. The authorization was done based on various factors, and after the successful comment, the stakeholders updated the blockchain. However, in this study, the integration of smart contracts was missing, leading to blockchain security issues. An integration of the supply chain along with counterfeit reporting was implemented using hyper ledger fabric by the Linux Foundation's *Abdulkader et al. (2019)*. One of the researches proposed by *Stopfer, Kaulen & Purfürst (2024)* referred to wood supply chain management using blockchain technology. Identification for the availability of drugs was suggested by *Huang, Wu & Long (2018)* in which legitimate users of the system uploaded traceable drug information. For the COVID-19 vaccine tracking, a system was indicated by *Antal et al. (2021)*. The complete monitoring was done based on an Ethereum blockchain. The system generates a QR code for every user who claims to obtain a vaccine. The user must prove his authenticity and register using the proper sign-on technique. The user's body temperature was requested to be submitted to the system at the time of the claim. The patient's identity

was verified before injecting the vaccine, and any probable reports related to the inappropriate consequences were reported in the blockchain. The integration of Cloud Computing with the supply chain is given in *Surucu-Balci, Iris & Balci (2024)*. *Abbas et al. (2020)* suggested that the identity of the pharmaceutical store providing the vaccine should be authorized. It offered an extra add-on to the system by integrating the authenticity identification of the pharmaceutical store. *Dehshiri & Amiri (2024)* implemented such solutions using Z-Numbers. *Singh, Dwivedi & Srivastava (2020)* suggested a blockchain model to identify the surrounding temperature where the drugs were carried and transferred from one place to another. The hyper ledger fabric used in this architecture records the timestamp and the temperature of the drugs transported from one location to another. It helped to track the location identification of the drug delivery. As per the review submitted by *Dietrich et al. (2021)*, almost all the mechanisms suggested by various models support only a simple manufacturing chain for the pharmaceutical industry.

A comparative study of several proposed ideas is presented in Table 1. The proposed architecture tries to improve all existing models with proper technology integration. A theoretical acceptance for various models is done without a real-time implementation for IoT-based systems like *Musamih et al. (2021)*. It is also worth mentioning that our system has enhanced security. Using blockchain in integration with Smart contracts improves the privacy policies for individual patients under consideration. As per the study given by *Musamih et al. (2021)*, once the authentication was done, there was no need for further control of any user permissions. *Alkhoori et al. (2021)*, also quoted a similar fact. The data availability in the cloud architecture represented in the study uses single-time authentication, making it challenging to manage the critical data. Non-fungible tokens are used in this research to ensure more safety and traceability. The unique nature of these tokens provides a distinguished transaction and uniqueness in blockchain commits. Unique identification is done with the help of these tokens, which allows the user to authenticate and maintain data integrity. *Arcenegui, Arjona & Baturone (2020)* used the self-authentication system for the IoT devices connected in the proposed study. This exploited the non-fungible tokens to ensure the security of the IoT device itself. The proposed research makes use of ESP-32-based devices along with the Ethereum blockchain. The authors in the proposed model (*Omar & Basir, 2020*) also suggested using non-fungible tokens for the pharmaceutical blockchain. This study used radio frequency identification and near-field communication to track the information. To follow the transaction procedure of the entire system, the authors tried to use the Ethereum blockchain along with smart contracts. No further evaluation and testing was done for the proposed architecture in the study. The scalability issue, along with the performance of the proposed model, was left over. A further comparison of different types of tokens was given in a study by *Westerkamp, Victor & Küpper (2020)*. The performance of the transactions taking place based on latency was presented in this study. The authors used the traceability of the goods based on the non-fungible tokens approach. Using blockchain, it is easy to identify the originating location for the manufactured goods.

**Table 1 Comparative study of state-of-art-methods.**

| State-of-the-art models | Blockchain | NFT token | IoT device | GPS sensor |
|---|---|---|---|---|
| *Chandan, John & Potdar (2023)* | Yes | No | No | No |
| *Kazancoglu et al. (2023)* | Yes | No | No | No |
| *Tsolakis et al. (2023)* | Yes | Yes | No | No |
| *Nanda, Panda & Dash (2023)* | Yes | No | Yes | Yes |
| *Hasan et al. (2023)* | Yes | No | No | No |
| *Chen et al. (2023)* | Yes | No | Yes | Yes |
| **Proposed** | Yes | Yes | Yes | Yes |

**Table 2 Some tasks performed by NFT *Turki et al. (2023)*.**

| NFT functions | Task performed |
|---|---|
| Balanceof( ) | Provide a total number of NFT tokens |
| Ownerof( ) | Identifies the user |
| Safetransferfrom( ) | Maintain exchange of safe transfer |
| Transferfrom( ) | Transfers ownership |
| Approve( ) | Approval of the use |
| Getapproved( ) | Ensure the approval |

# PROPOSED MODEL IN THE STUDY

This section comprises all the information related to the working model and the components. The infrastructure required to build the model needed for the supply chain in the drug industry is mentioned here. The complete architecture is divided into several layers, which comprise all the objectives required for the traceability of the drug in various instances. The architecture represented integrates multiple services and components needed to work the entire system. We try to split all the elements to ensure a clear and brief understanding of how the various layers work. Non-fungible tokens are used to identify the authenticity of the different IoT units involved in the study. Figure 1 represents the proposed model in this study.

The involvement of several actors inside this proposed architecture begins with the initialization by the SFDA. The SFDA provides the delivery of permission to execute the manufacturing of drugs to a manufacturer after a complete background check and validation. The Central Food and Drug Authority of Saudi Arabia provides licenses for all the users in the prescribed study. The central authority also manages the license renewal and the penalties. The manufacturer is responsible for creating quality drugs and shipping them to the distributors in proper locations. The distributors are responsible for receiving the drug and distributing it further to the small-scale pharmacy stores. The distributor collects drugs from the manufacturer and submits them to the pharmacy stores whenever required. The distributor further ensures that the vehicle transporting the drugs from the warehouse to a pharmacy store comprises all the facilities needed. As soon as the vehicle

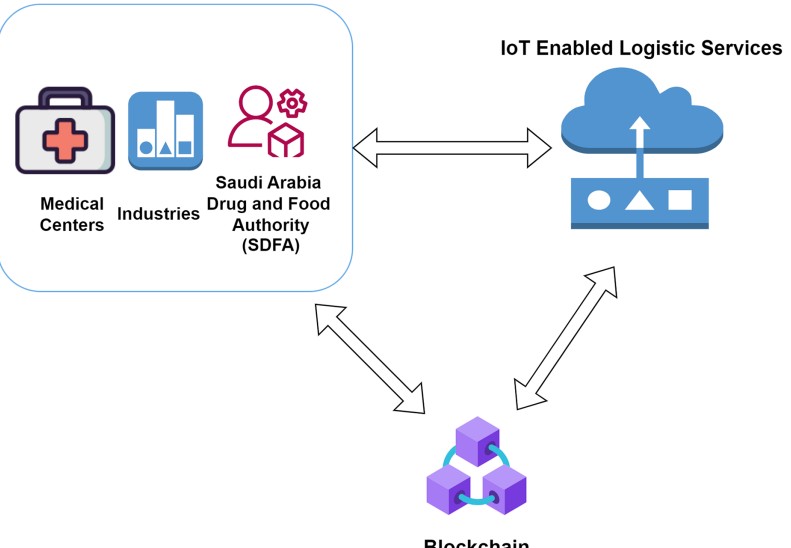

**Figure 1  Proposed architecture for Saudi Arabia.**     

interacts with the warehouse and the pharmacy store, the blockchain data is submitted along with the drug information to understand the IoT status for the temperature as well as the manufacturing details of the drug boxes. The vehicles have IoT services to provide proper information about the drugs carried at remote locations. The hospitals or pharmacy stores that demand the drugs use drugs to treat the patients and help them recover from their diseases. Before the delivery of the drugs at the hospital or pharmacy store, the members are supposed to check the drug's authenticity and commit to the final blockchain data block. Once the drug's validity is assured, the hospital or the pharmacy store executes the last blockchain to reflect the successful delivery of the drugs. The evaluation for the manufacturer and the distributor is done by the Saudi Arabian Food and Drug Authority (*AlQuadeib et al., 2020*).

## Components of the model

Several components are considered when building the entire system in the proposed architecture. This comprises various stakeholders along with the Internet of Things-based devices. The data storage, along with the interactive web-based application, collects information from multiple stakeholders. The users are managed with the help of identity access management using non-fungible tokens. One of the essential parts of the system is comprised of smart contracts that submit information to the blockchain. The components which are involved in this study are as follows:

### *Sensor-based devices/IoT enabled monitoring services*

The devices responsible for collecting data from various system parts and streaming them to the network are gathered in this category. Primarily, the devices containing sensitive information from the sensors related to the temperature and other physical conditions submit the data. GPS sensing, an essential part of transporting medicines and other drugs,

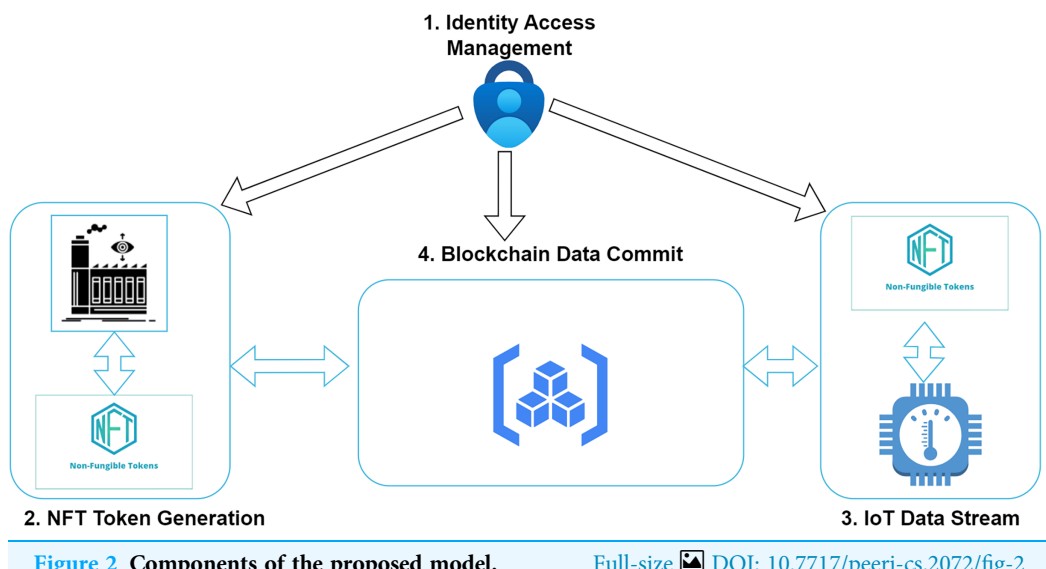

**Figure 2** Components of the proposed model.

is also controlled under this part. Blockchain access to the devices and their information is done with the help of non-fungible tokens (*Wang et al., 2019*). This, in return, contains the identity access management of these devices. The successful enrollment of the device MAC addresses and other physical features into the blockchain makes authenticity easier and preserves data privacy. The data collected by the sensors primarily comprises the temperature and humidity conditions. Once the data is connected and the temperature or any other physical feature exceeds the minimum threshold value, the drugs are marked as invalid to be used. Since the drugs are damaged, it is not kept inside the blockchain, and the GPS locations are sent to the blockchain where the drugs are coagulated. The data collected by the sensors is processed at local layers. Finally, the dressed-up data is submitted to the blockchain for final commit in the P2P network.

In this study, blockchain plays a vital role in the data to be committed in the P2P network. Ethereum is used in this case to provide privacy and efficiency for the blockchain network—exchanging cryptographic keys with the help of smart contracts at user-level authentication. The data storage, user authentication, and tempering of the information logs are essential parts of the entire blockchain platform presented in this study. Using non-fungible tokens and their integration into sensor-based IoT devices tends to produce a properly traceable and effective system. The transparency of the data, which is controlled and submitted to the blockchain ledger, contains the author's information along with the timestamp for submissions. Every transaction is tracked to make sure that the drugs do not get invalid. Smart contracts enable the stakeholders to provide valid and legitimate information.

The blockchain network comprises decentralized data storage, which travels between various nodes across the P2P network. The information inside the supply chain architecture proposed in this study consists of massive data. A single data centre is not

enough to hold onto the information. The distributed system manages data and files along with the transactions. The files stored as backups are more significant, which will be challenging to handle and upload to the blockchain platform. To improve the scalability of the system.

Figure 2 contains the primary components needed in this study. The main use for applying a non-fungible token is to trace the drug from the manufacturing level till the finally sold product. It is essential to realize the traceability of drug-related pharmaceutical transactions. A massive amount of data is submitted by various agencies manufacturing the drugs, including the suppliers, distributors, individual consumers, and agencies responsible for delivering the drugs, like hospitals or pharmaceutical dispensaries. The data from the transport unit also comprises critical information related to the temperature and humidity of the vehicle. This data has also been submitted on a significant level. In this study, we tried using two non-fungible tokens at two stages. These tokens are identified with a unique identity value for both the smart contracts at the two stages. The smart contract digital signature and the non-fungible token ID are associated with the specific object. These two values safeguard the secrecy and traceability from the manufacturing tail to the final delivery stage.

- At the beginning level, a non-fungible token is supplied by the SDAFA to the manufacturing unit corresponding to a specific production batch of the medicine. The NFT ensures that medicines can be traced depending on the behaviour of patients after consumption. Suppose the patients feel trouble after taking a specific medicine. In that case, the complete batch can be identified with the help of this non-fungible token, and the medicines related to these batch processing can be taken back from the entire market. Based on this medicinal NFT, tracking all the transactions related to the batch in which these medicines are processed is easy. Several owners and stakeholders are involved in the production of this cycle. The non-fungible token ensures that all use a specific token related to a production batch. The entire cycle from manufacturing to distribution is done with the help of this non-fungible token to ensure the traceability of the information related to medicinal behaviour.

- The vehicles which carry the medicine from one place to another are provided with a specific non-fungible token at the time of transport. The main idea for giving a non-fungible token to the vehicle is to locate the unique vehicle during the drug transport. The SDAFA will certify and authorise a car to ensure that the vehicle associated with the transport meets all the required standards. Also, the temperature and humidity during transportation are tracked by the non-fungible token value for the vehicle. The entire medicinal batch can be invalid if problems are associated with temperature or humid conditions. It might be harmful to the consumers later. The token ensures that the medicines which travel from one GPS value to another are tracked efficiently. The IoT devices connected to the vehicle during the travel ensure that the data from these sensors travels effectively to measure the proper handling of the drugs in the vehicle during transportation.

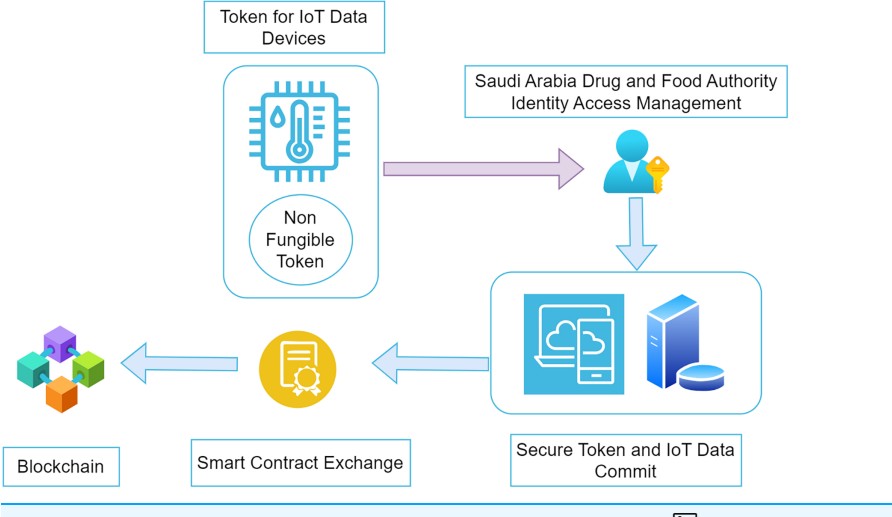

**Figure 3  Integration of NFT with blockchain.**     

### Executable modules

The registration for all the members and stakeholders is done with the support of SDAFA. A blockchain node address is submitted for each individual responsible for the complete supply chain. The centralised authority does the registration for the manufacturer of the drugs, the distributor of the processed drugs, and the hospital utilizing the drugs. All the verification for the authorities involved in the procedure is done with the help of SDAFA. The information is submitted inside the platform. The interaction between various entities involved in the system takes place with the help of a web-based application that can provide the stakeholders' inputs. The units responsible for manufacturing the drugs will be able to supply information related to the goods that are finally processed in the system. The shipment orders are calculated and submitted with the help of the distributors. The transport agents moving the medicines produced from one location to another successfully upload all the registered vehicle entries and other information. Information from sensor-based devices is sent from vehicles. Figure 3 represents the major parts of the system design in this research. All the core parts are divided into four different sub-units. Drug manufacturing is the task of the manufacturer, who has been certified and validated by the centralized authority of the Kingdom of Saudi Arabia. As the manufacturer receives approval related to the processing of a drug, it is responsible for creating the non-fungible token for the drug's production batch. The value of the non-fungible token is responsible for managing the identification and traceability of the drug across the entire lifecycle. At any instance in the complete supply chain, the NFT value is used by the various stakeholders to trace the batch for the drugs. As soon as an order arrives for delivery of medicines at any instance, the manufacturer identifies the non-fungible token value related to the order and associates it with the shipment. The destination address, valid status, non-fungible token ID and other information are stored in this order. A proper vehicle is identified, and the batch is given to the transport unit to take it to the next level.

Once the order is received and handed over to the transportation unit, another non-fungible token is generated for the vehicle. This token ensures that the data from the IoT

devices inside the car sends the information corresponding to that specific NFT value. The significant difference between the NFT value submitted in this transportation layer and its connectivity with the blockchain nodes is its connectivity. The environmental values, such as temperature and humidity for the medicine, are streamed across the IoT devices and are registered in the blockchain node. The distributor remains the primary stakeholder for the non-fungible token created in two words: the manufacturing and transport of the medicine. The accuracy of the trusted data sent from these devices for further actions is required to identify the validity of the drugs supplied. All the steps shown in Fig. 2 above cannot occur without proper authentication and authorization of the individual stakeholder. Security concerns are one of the main problems associated with the blockchain. The secure sharing of the non-fungible tokens across the blockchain is one of the significant factors that needs to be taken care of. Malicious access to the non-fungible tokens may be a hazard to the existing consent of the owners. Role-based access control allows smart contracts to handle transactions between stakeholders by providing proper authorization. Potential threats to the entire supply chain of drugs can be minimized with the help of non-fungible tokens in association with smart contracts. Any malicious activity related to data access can be identified in the transactions. Placing the complete setup ensures all the layers of the user's authenticity when using the system. The data storage for various entities in the proposed architecture occurs in the blockchain. It further enhances and modifies the level of security for the architecture presented in the system.

## IMPLEMENTATION

This study is organized in several ways, representing the creation and deployment of the system required for a proper supply chain in Saudi Arabia's pharmaceutical industry. The Saudi Arabian Food and Drug Authority is vital in identifying quality standards for drugs and their manufacturing. Several factors should be taken into consideration when designing a system. The primary stakeholder (SFADA) is responsible for providing the validation and verification of the industries that produce pharmaceutical drugs. On proper validation per the country's standards and international global medicinal standards, adequate login and access are provided to the stakeholders for the system. Manufacturers, distributors, hospitals, drug stores, pharmacies and warehouses are the premium users of the system. Access is provided based on identity and access management techniques. Complete identity access is developed based on verifying the stakeholders in the system. Once the identity access is completed, the system can be accessed at different levels depending on the stakeholder's involvement. The organizations responsible for completing a specific task will be able to access only the required features of the system. Meanwhile, users who are not legitimate to use the desired features will be able to access only their part of the system instead of the complete system. Several steps are involved in creating a secure blockchain, including smart contracts and system accessibility.

Blockchain implementation: In the initial phase, the manufacturers tend to offer a non-fungible token to the vehicle that will commute the finished products into the market. Token values are available to the batch producing a specific drug. During the transfer, physical quantities like temperature, humidity, *etc.*, are calculated and observed during the

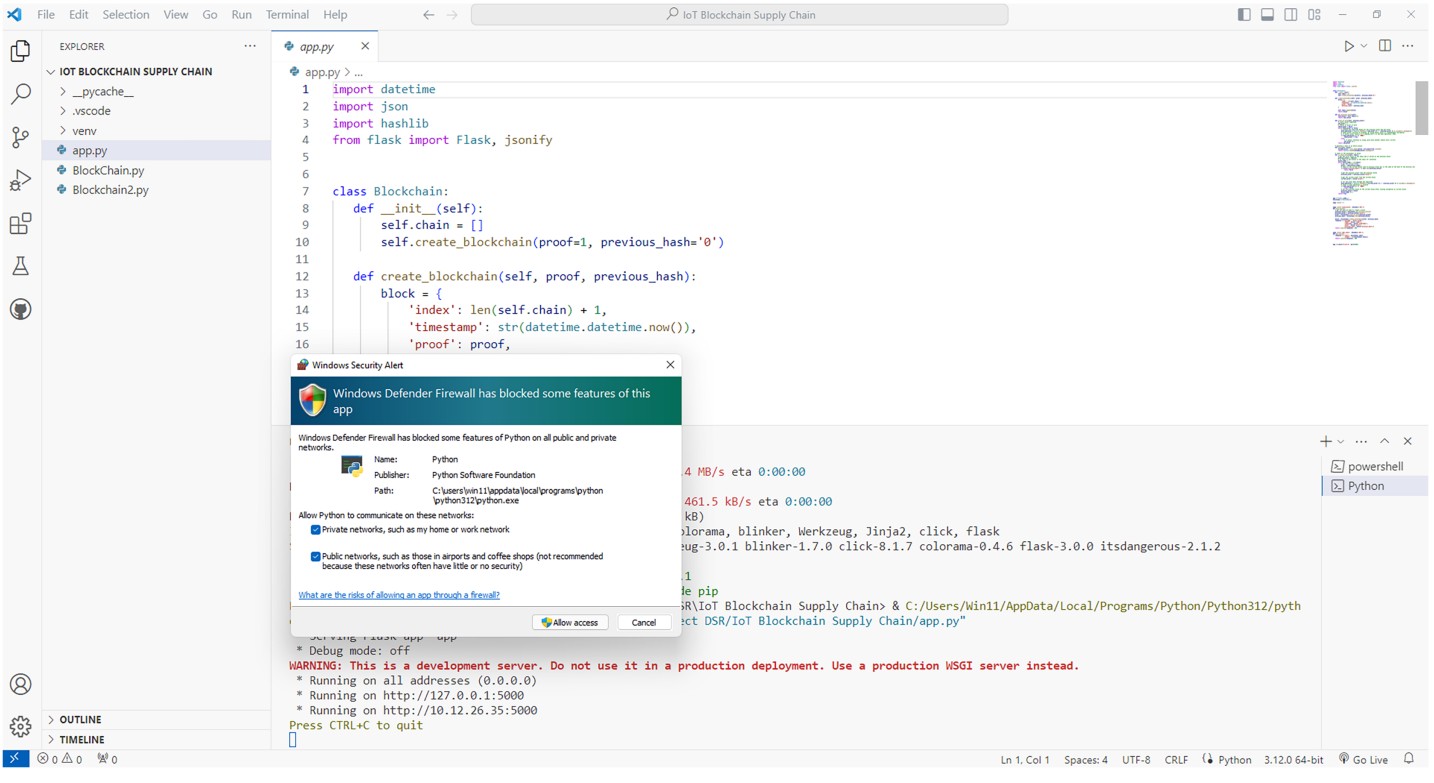

**Figure 4** Activation of blockchain in a local test network for the proposed model.     

transportation. These IoT devices can submit the information directly to the Ethereum hyper Ledger blockchain. Once the complete hyper-chain is created for a specific non-fungible token relative to a batch of produced goods, it is submitted, and a block is made at the P2P network. The integration is represented in Fig. 3

Figure 4 shows the creation of a blockchain in a test network. The blockchain and the exchange of smart contracts are activated on the local network.

In Fig. 5, the peer nodes inside the system access the blockchain network. These smart contracts are activated to transfer information related to non-fungible tokens from IoT devices. The devices are streamed the data from one location to another during the transportation of the pharmaceutical drugs. The smart contracts and blockchain comprise these NFT tokens' hash value. As represented in Fig. 5, the blockchain unit is created and deployed on the test network by integrating the smart contracts. Amongst the three phases of the system's working, creation of NFT tokens, deployment of blockchain and exchange of smart contracts are the most essential components. Every non-fungible token is associated with a smart contract for secure data exchange.

In Fig. 6, a smart contract related to the mind information from the NFT token created in the previous node is exchanged. Upon activation of the next node, its hash code is sent to the blockchain to maintain secrecy and privacy. This ensures that the information is exchanged in the proper note. The non-fungible tokens are entities responsible for finding

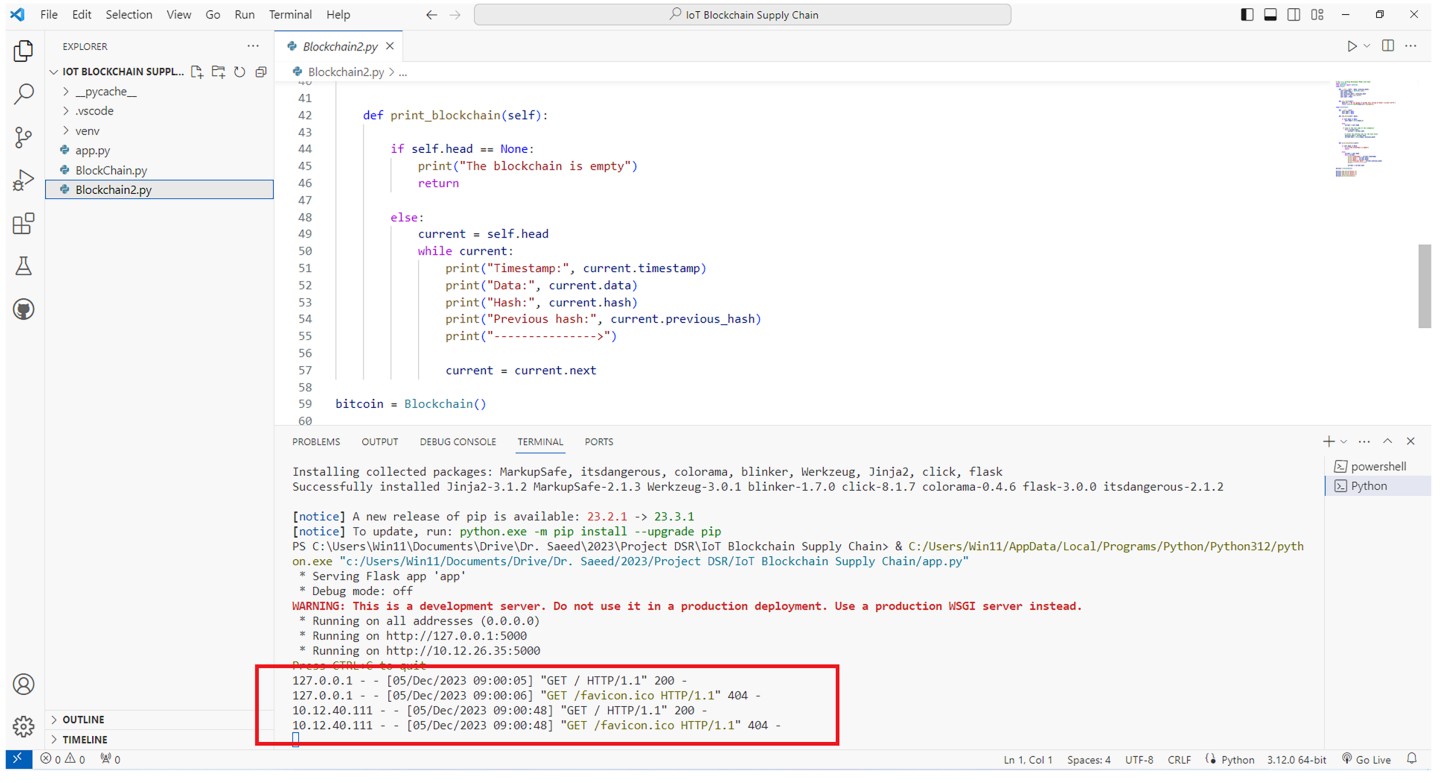

**Figure 5  Secure blockchain access in the test network for the proposed model.**

**Figure 6  Blockchain secure hash commit for the proposed model.**

data with the private key encryption system. All the information that streams from the vehicles taking the patches of medicines from one place to another is associated with the ERC721 NFT tokens. The manufacturers of the drugs have the authority to associate NFT tokens with the batch of the drugs produced at a specific time. These drugs are transported with the help of IoT-enabled vehicles in which the devices can sense physical conditions and submit information related to the NFT tokens. The tokens are responsible for exchanging information on the blockchain with the help of proper information and

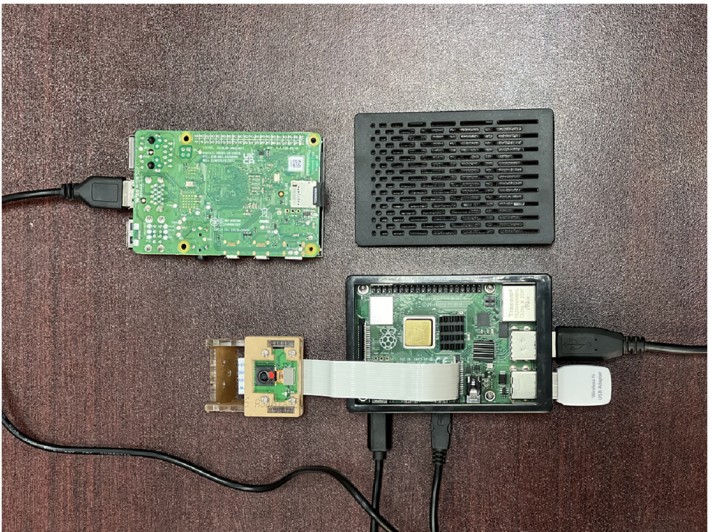

**Figure 7  IoT device integration for the proposed model using Raspberry PI 4b model.**

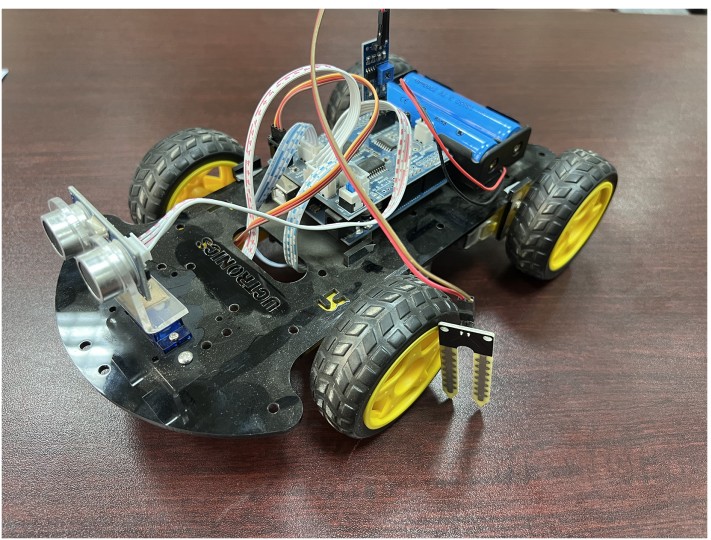

**Figure 8  Demonstrative IoT enabled transportation vehicle.**

support system. The secrecy and privacy are maintained based on which the final data is pushed to the blockchain node. IoT devices are connected in vehicles transporting medicinal drugs from one place to another. These IoT-enabled devices relate to a small computational unit such as Raspberry PI 4B or Arduino UNO.

Figure 7 represents a typical integration of Raspberry PI 4b, along with several integrated circuits to measure the physical conditions in the vehicle. The unit relates to IoT-enabled sensors that can stream data from various locations. The small computational unit contains a Wi-Fi-enabled adapter with power sockets and is connected to another communicating small computational unit.

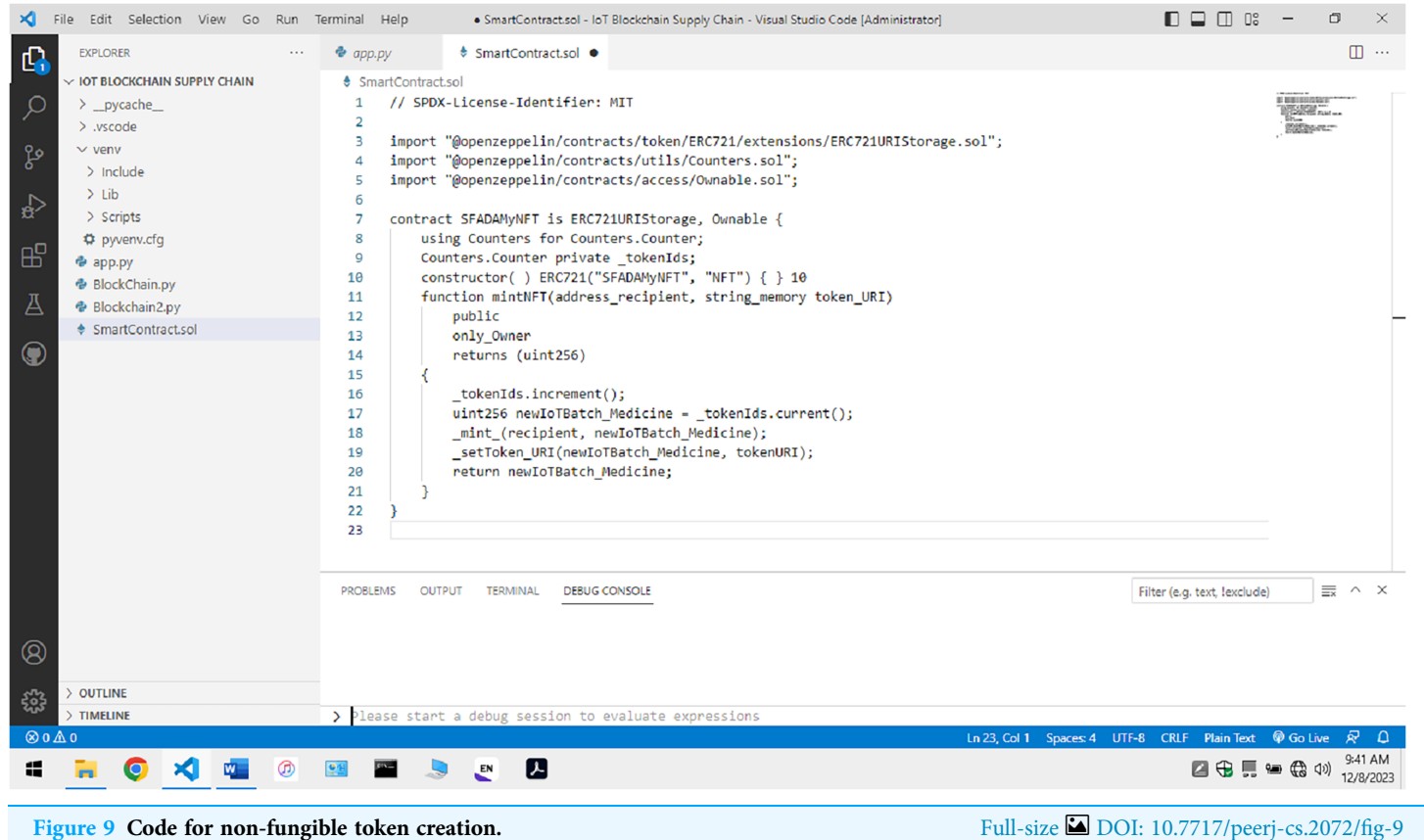

**Figure 9  Code for non-fungible token creation.**

As shown in Fig. 8, a vehicle comprises a small computational unit integrated with a temperature sensor and humidity measurement sensor. Several other IoT-enabled devices can be coupled with the computational unit to enable data transfer to the blockchain. The vehicle contains an associative non-fungible token integrated with the batch of the processed drugs to be transported from one place to another. The small computational unit enables the system's connectivity with the associated IoT devices. Data from these devices flows from sensors towards the centralized computational blockchain node. The sensors can collect information on temperature and GPS. They are capable of streaming it continuously. All the information is associated with an NFT token to ensure secrecy.

Non-fungible token generation: The creation of non-fungible tokens for the safe transfer of information and the secure exchange of smart contracts occurs. As depicted in Fig. 3, the non-fungible tokens were generated by the manufacturers who were authorised to create batches of finalised drugs. During the transfer fraud, these drugs are transferred from one location to another, and the NFT values are exchanged from the manufacturer to the distributor. Based on the stakeholder NFT value, a safe comet on the blockchain depends on natural factors such as temperature, humidity, pressure, *etc.* The data from the sensor devices flows across the blockchain along with the NFT token value to ensure secrecy and privacy. Several functions are required for the smooth functioning of the non-

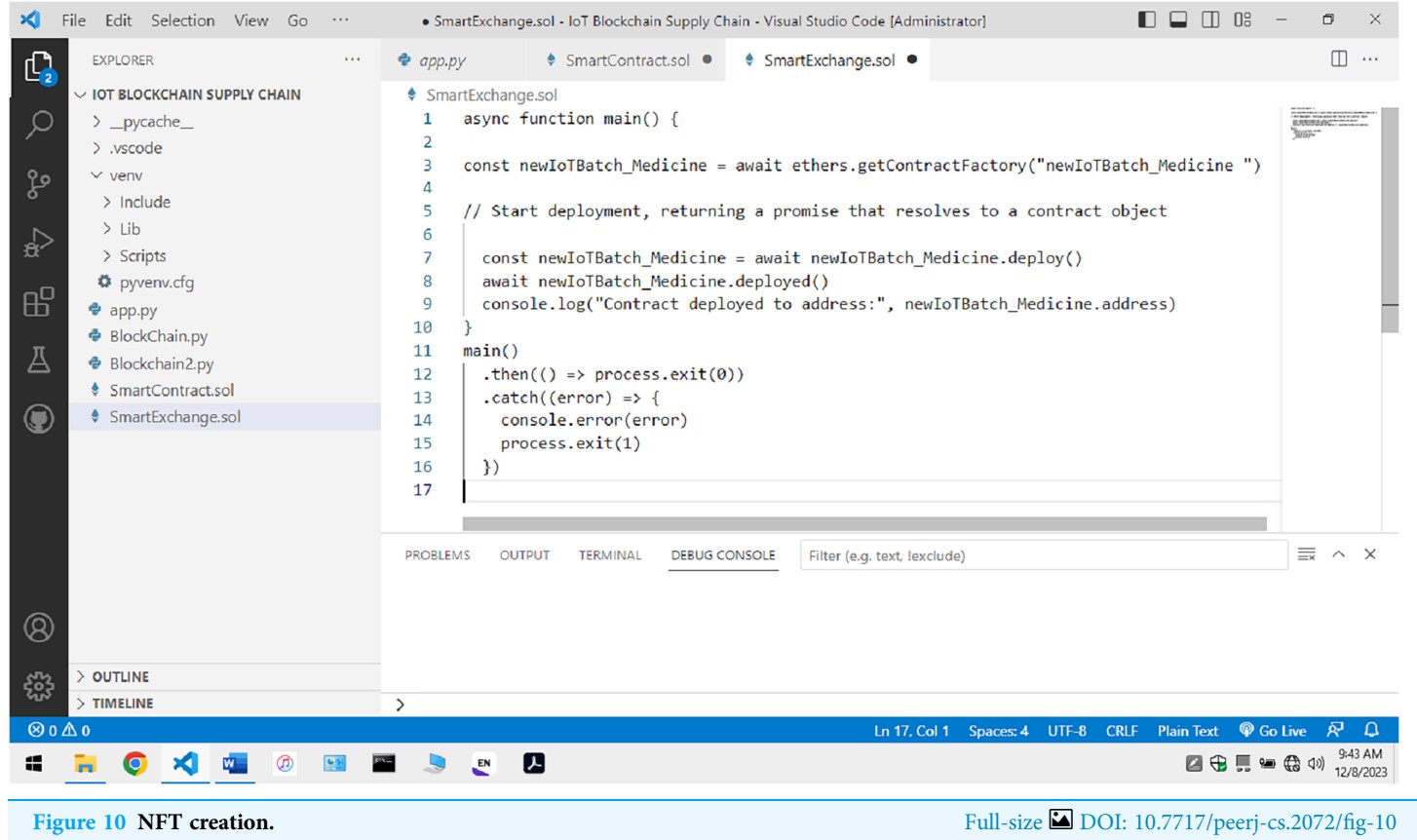

**Figure 10 NFT creation.**

fungible token values. These functions are implemented to ensure that the working of non-fungible tokens is helpful and consecutively takes place. Several functions are implemented but are not necessary from this study's point of view. The model presented in this research tends to implement the functions which are described in the table below:

The functions depicted in Table 2 are implemented in the prototype model and created for testing at the local test network. The prototype model comprises the creation of non-fungible tokens to ensure secure transmission and transaction. The tokens produced by the manufacturer are used to submit information from the vehicle's sensor devices during transportation. The ERC721 standard of non-fungible token generation ensures a two-way secrecy protocol is implemented in the architecture.

As shown in Fig. 9, a non-fungible token is created using the ERC 721 standard for the transaction to occur. The manufacturer submits information about the batch of pharmaceutical drugs made and this token value. The token value ensures the identification and traceability of the batch of drugs created at that instance. It is connected with the timestamp values to ensure that a proper tracing of the Timeline related to medicine's life cycle is maintained across the complete transaction.

Once the non-fungible token is created and deployed along with the other information, the sensor-related data is sent to the blockchain. Figure 10 ensures data submission across

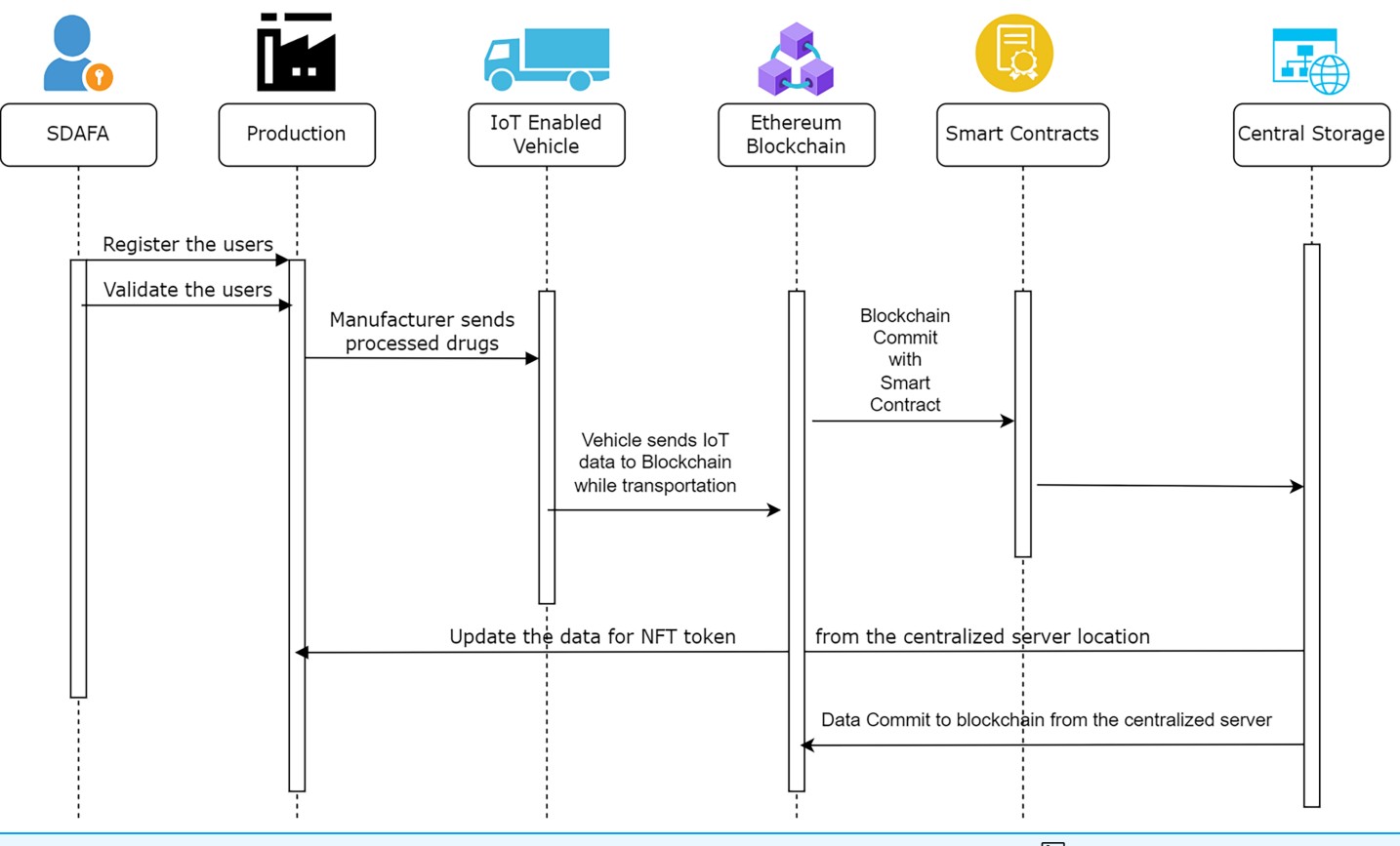

**Figure 11 Sequence diagram for the proposed model.**

the blockchain takes place with the Association of Non-fungible Tokens to ensure secrecy and privacy. The information submitted to the node is saved once the NFT security credentials of the sensor devices match. The identification of the information is done with the help of the algorithm below:

INPUT: NFT Token Value
   OUTPUT: Sensor Data Values
   getBatchDetail{}
   for all NFT values
   GET ProductionDetails( )
   CheckProductionDetails( )
   getParameterValues( )
   if ParameterValues ¡= ThresholdValues
   Commit the value to node
   Else CommitValue = = FALSE
   ProductionDetails= ProductionDetails + NFT_Value + AllSensorData
   End FOR
   Return ProductionDetails

The algorithm shown above refers to commenting the data into the blockchain node depending upon the identification of the sensor data. Once the data accessed relative to a non-fungible token value is found at par with the threshold values of the required parameters, it is committed inside the P2P nodes in the blockchain.

The sequence diagram in Fig. 11 summarizes the complete process of the supply chain proposed in this study. Various stakeholders inside the system are represented by the core functions performed by each agency. The Saudi Arabian Food and Drug Agency provides registrations to authorised manufacturers and distributors. The production occurs at the manufacturing site, and the vehicle is transported after the manufacturer creates the non-fungible token. A generic web application is completed to ensure that the data flowing from one location to another is made available for identity access, depending upon the individual stakeholder.

## DISCUSSION

One of the major concerns that is expected to appear in this article is security and privacy. The proposed system architecture uses the non-fungible token associated with the smart contracts. This study refers to a secure mechanism responsible for submitting the data from the batch of processed drugs travelling from one location to another. Several factors are taken into consideration before designing this model:

A non-fungible token is created with the help of the ERC 721 standard (*Turki et al., 2023*). It ensures the careful verification and monitoring of the information related to a specific NFT value. Every stakeholder can provide the steps in the complete supply chain related to the task performed. Ethereum is used to deploy blockchain services with a broader range of applications involved. The malicious users are unethical operators who will not investigate any issues related to the transmission of information from the sensor devices to the blockchain. We have deployed the testing in the test network with the help of several open-source tools to identify the vulnerability. Wireshark and Smart Check further provide a proper analysis of the vulnerability check. Several vulnerabilities are detected in this case. After carefully investigating the vulnerability features during the exposure analysis phase, a smart contract is deployed in the test network to avoid the vulnerabilities. The authorization and authentication for the proposed architecture become more integral and safe. All the transactions in the above-proposed model react with the eligibility of non-fungible tokens and smart contracts. Figure 11 shows the test implementation of a smart contract approval. In the P2P network, once the smart contract is approved, information exchange occurs between the two parties. Figure 12 represents a test network where smart contracts are exchanged for demonstrative purposes. Data from vehicles transferring drugs from one place to another is synchronized with the help of non-fungible tokens, but at the same time, it is submitted to the blockchain. Data is saved in the blockchain on the successful exchange of tokens between the parties. This proves the data's integrity and maintains the information's privacy. The Ethereum blockchain efficiently uses information submitted by the user nodes. All the parties involved in the transaction contain their own secure NFT token values and authorization credentials using identity access management. After the successful verification of the authenticity of the user, the final data is committed.

**Figure 12 Smart contract exchange between two nodes in the blockchain.**

The stakeholders are identified based on the addresses provided in the blockchain. The vehicles transporting the drugs are equipped with non-fungible token values, which allows tracking the drugs and securely and confidentially submitting the physical factors. The authorization in the blockchain network takes place without any third-party intervention. The transparency of the blockchain ensures that every stakeholder has access to the specific entities involved. The data generated in the blockchain consists of several fields explained below:

- Index: the index value comprises the index of the block processed.
- Message: the data field for the messages comprises messages submitted along with the blockchain node.
- Hash: contains information related to old nodes in the blockchain.
- Proof: it contains the numeric value for the proof in the blockchain node.
- Timestamp: comprises all the information related to the time and the date for the block commit.

The information closely tracked in the blockchain node is integrated with uploading data from various nodes in respective fields arriving from the stakeholders. All the blockchain entries are committed to the network with these data to ensure their integrity.

NFT Date is generated for the stakeholders in the proposed model for the supply chain. The following data specifications are developed in this study:

- Balance: provide the total number of NFT tokens generated for the transmission.
- Ownership: represents the ownership domain for any specific stakeholder in the supply chain.
- Transform: provides the information about the transformation of data exchange from one node to another in the prescribed supply chain.
- Approve: represents the information approved by the stakeholder in the hierarchy for transmitting the medicinal drug supply chain.

## Data in the smart contract exchange

The exchange of smart contracts between two entities at the time of blockchain commit is represented by Fig. 12. The data presented in this case comprised of the following entities:

- Organization: represents the sector of the stakeholder responsible for the exchange of smart contracts. It contains the information related to the first party, and the second party will exchange the data.
- Approvals: contains the flag values for approval of the data between the two organizations exchanging a smart contract.
- Package ID: represents the data package value responsible for tracking the information flowing from one node to another.
- Peer address: contains information about the node participating in the smart contract exchange over the P2P network.

All information that travels across the blockchain network is secure by vital mathematical functions and cryptographically signed. The proper signature of the mathematical functions ensures that the data is preserved safely and securely. The transactions in the blockchain network are associated with a specific NFT value and smart contract credentials. All the information is recorded in the form of log entries. No type of denial is expected in the transactions, ensuring the complete safety and security of the information.

## Future directions

Blockchain technology is spreading across the world due to its transparency. Several organizations have started adapting blockchain in their supply chain integration. The prescribed model is a complete solution for a substantial pharmaceutical drug supply chain in Saudi Arabia. Several issues can be addressed soon. The proposed model contains technical updates on developing an IoT-enabled secure blockchain. However, there is a scope for integrating artificial intelligence into this field. Several algorithms from machine learning can be combined to provide artificial intelligence in the supply chain sector. The article prescribes using NFT tokens for security amongst different stakeholders. Several advanced non-fungible tokens can be integrated to provide more secure transactions. The

data processing with the help of smart exchange can be enhanced further. Due to the recent implementation of blockchain, security is still a big concern. There is still a vast scope for improving security in the blockchain domain with the help of sophisticated algorithms. The researchers are invited to develop and investigate the use of such algorithms to provide more accuracy in P2P network transactions.

## CONCLUSION

This research focuses on developing a model for the supply chain of medicinal drugs in Saudi Arabia. Non-fungible tokens further make it safe and private. Using IoT devices in coordination with NFT tokens makes it possible for the stakeholders in the proposed model to ensure the system's security. Non-fungible tokens and the Ethereum blockchain network provide information about the drugs that is retrieved at every instance. The vehicles that carry the drugs from one place to another are equipped with IoT devices to share physical data, such as temperature, humidity, *etc.*, for the drugs. The traceability of the medicines is possible with the help of the NFT values along with their expiry or invalid nature. A smart computational device enabled with temperature and humidity sensors is used in the test study of the model. The information is submitted with the help of the ERC 721 NFT standard. We also implemented the protocols for exchanging information from the NFT to the blockchain network. To add further security, smart contracts are developed in the test network for the information flowing from the IoT-enabled vehicles to the blockchain. We also discussed the security issues and their possible solution for the implemented prototype of the model proposed in this study. As a future work, machine learning can be beneficial for providing an edge to the proposed model. We aim to integrate artificial intelligence to predict and trace drugs in Saudi Arabia.

### Funding
This work was supported by the Deanship of Scientific Research at Shaqra University through the project number SU-ANN-2023040. The funders had no role in study design, data collection and analysis, decision to publish, or preparation of the manuscript.

### Grant Disclosures
The following grant information was disclosed by the authors:
Deanship of Scientific Research at Shaqra University: SU-ANN-2023040.

### Competing Interests
The authors declare that they have no competing interests.

### Author Contributions
- Saeed M Alshahrani conceived and designed the experiments, performed the experiments, analyzed the data, performed the computation work, prepared figures and/or tables, authored or reviewed drafts of the article, and approved the final draft.

## Data Availability

Amongst the three phases of the working of the system, creation of NFT tokens, deployment of block chain and exchange of smart contracts are the most important components. The code available in the Supplemental Files is used to create an NFT Token and blockchain base network for testing. It also includes the Smart Contract Exchange for the system proposed.

## Supplemental Information

Supplemental information for this article can be found online at http://dx.doi.org/10.7717/peerj-cs.2072#supplemental-information.

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
