# Peer review of "Enabling blockchain for Saudi Arabia drug supply chain using Internet of Things (IoT)"

_PeerJ Computer Science, doi:10.7717/peerj-cs.2072_

## Round 0.1 · original submission · Major Revisions

Dear Dr. Alshahrani,

Reviewers find merit to your manuscript, however, they suggested a significant revision. You must address all the reviewers' comments and suggestions and submit a revision. The revised manuscript will be subjected to a 2nd round of review. Good luck.

**Language Note:** The review process has identified that the English language must be improved. PeerJ can provide language editing services - please contact us at [email protected] for pricing (be sure to provide your manuscript number and title). Alternatively, you should make your own arrangements to improve the language quality and provide details in your response letter. – PeerJ Staff

Reviewer 1 ·

Basic reporting

This manuscript introduces a token-based blockchain model for a reliable drug supply chain through IoT.

- The paper is easy to follow.

- Recent paper published in 2023 in not available in literaure survey. Though, few papers are discussed in sub-section 1.1

- Name of the components in the figure 1 and 2 (like blockchain, network layer IoT data transfer) should be more clear.

- There must be a separate sub-section as "Our Contributions" in Introduction section. Also, it should highlight 4-5 main contributions in terms of novelty.

- The remaining portion of the paper should be in the manuscript as the last paragraph in the Introduction section.

Experimental design

- Code snippets or screenshot should be placed in a GitHub repository along with the implementation code and revised section-4 accordingly.

- Compare your technique with recent state-of-the-art works/models.

- Show evaluation results

- Perform Ablation study.

- Discussed data used in this work in detail as a separate sub-section.

Validity of the findings

- Future directions of research as a separate sub-section.

- Experimental details or parameters should be placed in a table separately.

- Results should be highlighted in bold and placed in a table.

Additional comments

Major revision

Reviewer 2 ·

Basic reporting

The paper idea is valid and added knowledge to the literature. It is related to the journal scope. Overall, the paper is an accepted with major changes. The limitation of the paper lies in paper structures and its presentation as it has to be improved.
1. The introduction section isn't well structure. It has to be rewritten using the following paragraphs: general information, research problems, the gaps, the proposed approaches, the results, paper structure. 2. You have to briefly describe the proposed approach in Introduction Section.
3. It will be a good idea of the authors can add a new table at the end of the second section related works to summarize and compare the existing approaches. The table should include the following columns: reference no, published year, approach name, advantages and disadvantages.
4. The paper has to be proofread.

Experimental design

The paper idea is valid and added knowledge to the literature. It is related to the journal scope. However, the author has to improve the following:
1. Add an algorithm to explain his contribution methodology.
2. Update figure 11. It isn't an activity diagram while it is a sequence diagram.

Validity of the findings

The paper idea is valid and added knowledge to the literature.

Reviewer 3 ·

Basic reporting

please check below

Experimental design

please check below

Validity of the findings

please check below

Additional comments

The authors have highlighted a very good model for managing the supply chain of medicines across the kingdom of Saudi Arabia. The model reflects technique related to management of organizing and tracking the complete system of drug manufacturing in the kingdom. There is a strong need for similar system for managing the supply chain of medicines not only in the kingdom of Saudi Arabia but across the world. The manuscript raised a great model by using block chain for the management of the supply chain. However, there are several updates that are required to make the proposed model more efficient:

1. The language of the manuscript should be improved and enhanced to of publication level.
2. The authors are advised to explain Ethereum block chain in the manuscript.
3. The use of IOT devices is done in the system, however, the authors must explain the technical specification of the IOT devices used.
4. The algorithm proposed for the identification of information in line number 525 is not defined correctly. The authors are required to update the format of the algorithm for better understanding.
5. A quick introduction to ERC 721 standard is also needed for the readers to acquire proper attention to words the proposed model.
6. The authors are also expected to use more recent references in their literature review.
7. The introductory session is well defined, however, it can be concise for better understanding. The authors can update the introduction to make it more effective.
8. The author must ensure that all the figures and the tables are numbered and cited in the reference properly.

Reviewer 4 ·

Basic reporting

In general terms, study offers a different perspective on IoT technology. However, my suggestions below should be evaluated throughout entire study;
1) Language of article should be a little more professional. English is insufficient.
2) Figure 1 looks very blurry, I don't know if this is related to loading, but it becomes meaningless for reader.
3) For experimental results, striking results should be made more evident with a table or figure. It is not clear exactly how article differs from other examples in literature in this form.

In addition to above comments, basis on which calculations and operations are made must be proven by giving formulas or references to formulas used.

Experimental design

.

Validity of the findings

.

Additional comments

Unfortunately keyword 'the' is used everywhere in the work, which affects integrity of paper and prevents it from being professional.

---

## Round 0.2 · accepted · Accept

I am pleased to inform you that your paper has been accepted for publication in PeerJ Computer Science. Your manuscript has undergone rigorous peer review, and I am delighted to say that it has been met with high praise from our reviewers and editorial team. Your research makes a significant contribution to the field, and we believe it will be of great interest to our readership. On behalf of the editorial board, I extend our warmest congratulations to you.

Reviewer 1 ·

Basic reporting

All comments are addressed properly.

Experimental design

All comments are addressed properly.

Validity of the findings

All comments are addressed properly.

Additional comments

Accepted in the present form.

Reviewer 4 ·

Basic reporting

When I reviewed the article, I saw that the authors made the corrections I requested. The work is suitable for printing in its current state.

Experimental design

When I reviewed the article, I saw that the authors made the corrections I requested. The work is suitable for printing in its current state.

Validity of the findings

When I reviewed the article, I saw that the authors made the corrections I requested. The work is suitable for printing in its current state.

Additional comments

When I reviewed the article, I saw that the authors made the corrections I requested. The work is suitable for printing in its current state.